# Impact of Various Ration Energy Levels on the Slaughtering Performance, Carcass Characteristics, and Meat Qualities of Honghe Yellow Cattle

**DOI:** 10.3390/foods13091316

**Published:** 2024-04-25

**Authors:** Lin Han, Ye Yu, Runqi Fu, Binlong Fu, Huan Gao, Zhe Li, Daihua Liu, Jing Leng

**Affiliations:** 1Key Laboratory of Animal Nutrition and Feed Science of Yunnan Province, Yunnan Agricultural University, Kunming 650201, China; 19286920591@163.com (L.H.); yy09091823@163.com (Y.Y.); fandrunqi@163.com (R.F.); binlongfu@126.com (B.F.); gaohuanhhh@163.com (H.G.); 2Faculty of Animal Science and Technology, Yunnan Agricultural University, Kunming 650201, China; qq2408731570@163.com; 3Animal Husbandry Office of Agricultural Comprehensive Service Center in Jin’an Town, Gucheng District, Lijiang 674104, China; 15812218358@163.com

**Keywords:** Honghe yellow cattle, energy level, slaughter performance, meat quality

## Abstract

Consumers are increasing their daily demand for beef and are becoming more discerning about its nutritional quality and flavor. The present objective was to evaluate how the ration energy content (combined net energy, Nemf) impacts the slaughter performance, carcass characteristics, and meat qualities of Honghe yellow cattle raised in confinement. Fifteen male Honghe yellow cattle were divided into three groups based on a one-way design: a low-energy group (LEG, 3.72 MJ/kg), a medium-energy group (MEG, 4.52 MJ/kg), and a high-energy group (HEG, 5.32 MJ/kg). After a period of 70 days on these treatments, the animals were slaughtered and their slaughter performance was determined, and the longissimus dorsi muscle (LD) and biceps femoris (BF) muscles were gathered to evaluate meat quality and composition. Increasing the dietary energy concentration led to marked improvements (*p* < 0.05) in the live weight before slaughter (LWBS), weight of carcass, backfat thickness, and loin muscle area. HEG also improved the yield of high-grade commercial cuts (13.47% vs. 10.39%) (*p* < 0.05). However, meat quality traits were not affected by treatment except for shear force, which was affected by dietary energy. A significant improvement (*p* < 0.05) in the intramuscular fat (IMF) content was observed in the HEG. Little effect on the amino acid profile was observed (*p* > 0.05), except for a tendency (*p* = 0.06) to increase the histidine concentration in the BF muscle. Increasing dietary energy also reduced C22:6n-3 and saturated fatty acids (SFAs) and enhanced C18:1 cis-9 and monounsaturated fatty acids (MUFAs, *p* < 0.05). Those results revealed that increasing energy levels of diets could enhance slaughter traits and affect the meat quality and fatty acid composition of different muscle tissues of Honghe yellow cattle. These results contribute to the theoretical foundation to formulate nutritional standards and design feed formulas for the Honghe yellow cattle.

## 1. Introduction

Insufficient beef production in China has led to beef and beef products being imported from countries such as Brazil, New Zealand, and Argentina to meet China’s beef demand [1]. In China, there is a rich resource of cattle breeds, the largest number of cattle breeds in the world, including fifty-three local breeds of cattle, seven breeds of cattle bred in China, and thirteen imported breeds of cattle [2]. Honghe yellow cattle are an important local breed of yellow cattle in Yunnan Province, China. In 1981, they were incorporated into the Honghe Hani and Yi Autonomous Prefecture of China’s Records of Livestock and Poultry Breeds [3]. Located in the Mourning Mountains, a mountainous region in the southwestern part of China’s Yunnan-Guizhou Plateau, Honghe yellow cattle are adapted for grazing roughage on steep slopes (at altitudes ranging from 2000 to 3166 m) [4]. And their advantages are that they are well adapted to harsh environmental conditions, have high resistance and fertility to parasites and diseases, and have good roughage/heat tolerance and adaptability. However, the disadvantages of Honghe yellow cattle are their small size, slow growth rate, low productivity, and low average slaughter rate (45.6%) [4].

As local Chinese yellow cattle have many benefits, improving their productivity and meat quality is essential for meeting consumer demand. The primary cause of low productivity in fattening cattle in developing countries is typically the use of corn stalks or other low-quality crop byproducts for feed supply and feeding techniques in the production of cattle. Low-quality feeds impact the consumption, processing, and energy levels of cattle, ultimately leading to a decrease in their overall performance [5,6]. Compared to cattle fed grass, cattle fed concentrate had more flavor, juiciness, and tenderness than cattle fed grass because they had less connective tissue and higher concentrations of monounsaturated fatty acids (MUFA) in their intramuscular fat (IMF) [7]. Moreover, consumers are increasingly discerning regarding the nutritional quality and flavor of beef. Consequently, due to the demands of the market today, techniques such as nutrition manipulation have been developed to increase the quality and quantity of beef. The energy concentration of rations plays an essential role in slaughter performance, carcass quality, and meat quality traits [8,9,10].

Meat quality, carcass quality, and animal development performance are all impacted by dietary energy levels, as research has shown [11,12,13,14]. Several studies have demonstrated that higher dietary energy levels enhanced the dressing percentage and backfat thickness [15,16,17], meat cut weight [18,19], meat protein content, IMF content [20,21], eye muscle area [16,19,22], fatty acid contents [23,24], and amino acid content [25] but reduced the shoring force of the muscles [26]. Previous research in various beef cattle breeds, including Angus [11], Angus × Holstein-Friesian, Angus × Gelbvieh, Angus × Limousin cattle [12,13,14], and F1 Angus × Chinese Xiangxi yellow cattle [15] have demonstrated a positive correlation between dietary energy and carcass traits and meat quality. Researchers conducted a 300-day feeding trial with Xiangxi yellow cattle as test animals and showed that cattle fed corn had dramatically increased IMF content and a* values, but reduced muscle moisture, cooking loss, shear loss, and drip loss compared to cattle fed primarily silage [27]. However, the results of these studies are not applicable to the production practices of Honghe yellow cattle because of the different dietary energy requirements of different breeds and growth stages of cattle. Additionally, there are few studies on the effects of dietary energy levels on Honghe yellow cattle. The objective of this research was to determine how different levels of dietary net energy affect the slaughter performance, carcass characteristics, and meat quality of Honghe yellow cattle. As such, the current study’s purpose was to explore fattening performance and meat quality features under various dietary energy levels in order to serve as a reference for healthy and efficient fattening in rural parts of southwestern China under full stall-feeding conditions.

## 2. Materials and Methods

### 2.1. Animal Ethics

This research was conducted in the light of the recommendations of the local Institutional Animal Care and Use Committee, the Institutional Management Committee, and the Laboratory Animal Ethics Committee (Approval No. YNAU20180020). This study was conducted from May 2020 to July 2020 in Yisa town, Honghe County, Honghe Prefecture, Yunnan Province, China (103 latitude and 23 longitude).

### 2.2. Honghe Yellow Cattle, Trial Design and Diets

A total of 15 Honghe yellow cattle (2 years old and 259.42 ± 23.87 kg in body weight (BW)) were separated and distributed randomly into one of three different groups, and each cattle was housed in an individual column (2.5 m × 4 m) with a one-way ANOVA design. The grouped cattle were randomly distributed to one of three rations with a comprehensive net energy (Nemf) level of 5.30 (low) MJ/kg, 5.60 (medium) MJ/kg, or 5.90 (high) MJ/kg, and a dietary protein level of 11.73%. Base diets were formulated for the breeding beef cattle under the study according to the Chinese Beef Cattle Feeding Standard (NY/T 815-2004) [28] to ensure that the nutrient levels in the diets met or exceeded the recommended levels. Moreover, the Nemf group’s ratio was formulated based on the nutritional needs of 260 kg of beef cattle in the finishing stage with an average daily gain (ADG) of 800 g. Compared to those of the Nemf group, the diet energy levels of the high-energy group (HEG) and low-energy group (LEG) were altered by 0.3 MJ/kg. Table 1 lists the diet compositions and chemical compositions of the experimental rations. The experiment was carried out over 70 days, which included 10 days of an adaptation phase and 70 days of data collection. During the experimental period, the diets were supplied twice a day at 7:00 and 19:00 in the form of a complete mix with free access to water.

### 2.3. Slaughter Parameter Measurements

At the termination of the experiment, all the cattle were then slaughtered using commercial methods. Briefly, preslaughter live weights (LWBS) of the cattle were recorded after 12 h of fasting, after which the cattle were slaughtered. We measured the backfat thickness, carcass weight, and dressing percentage after bloodletting, peeling, and removing hair, hooves, heads, tails, and viscera. Moreover, the noncarcass parts of the heart, liver, spleen, lungs, kidney, total stomach, small intestine, and large intestine were removed and weighed. When the meat was separated and deboned, the carcass meat yield, slaughter rate, and meat-to-bone ratio were computed. The area of the longissimus dorsi muscle (LD) and fat thickness were calculated between the 12th and 13th LD ribs using transparent sulfuric acid paper and a fat-measuring meter, respectively. The commercial meat cuts were dissected, named, and weighed after trimming following the Chinese standard for beef carcasses and cuts (GB/T 27643–2011) [29]. Among the most commonly purchased and cost-effective meat choices in Chinese markets, the highrib, ribeye, tenderloin, and striploin were considered top-grade cuts, while the brisket, thick flank, shoulder, outside flat, topside, eye round, and rump were considered medium-top-grade cuts. As a result, the top- and medium-top-grade cuts were calculated to determine their yields. Samples of approximately 800 g of the LD at the 12–13th rib of cattle and the biceps femoris (BF) muscles were quickly removed from the left side of the carcass, sealed in a sterile vacuum package, and refrigerated at 4 °C for assessing meat quality, and taken to Yunnan Agricultural University for analysis.

### 2.4. Meat Quality Measurement

The pH values of the LD and BF muscles were measured at 45 min and 24 h postmortem using a Magnetics PHBJ-260 portable pH meter (INESA Scientific Instrument Co., Ltd, Shanghai, China). Meat color determination was performed according to the methods of previous studies [30]. LD and BF muscle samples (3 cm × 3.5 cm × 2 cm) were taken within 2 h after slaughter and measured with a modified WW-ZA strain gauge (Nanjing Dongmai Technology Instrument Co., Ltd., Nanjing, China) without a measuring limit. The mixture was placed on a pressure test bench to maintain a pressure of 35 kg for 5 min, after which the meat sample was removed and weighed. The following equation was used: drip loss (%) = sample weight before pressing (g) − sample weight after pressing (g)]/sample weight before pressing (g) × 100%. The water-holding capacity was calculated according to the methodology described by Li et al. (2014) [15]. A sample of LD and BF muscles was heated in an 80 °C water bath to reach 70 °C core temperature, and the cooking loss was expressed as a percentage of weight loss before and after heating. A C-LM3B texture analyzer (Beijing Tianxiang Feiyu Instrument Co., Ltd., Nanjing, China) was used to measure the Warner-Bratzler shear force (expressed in kgf) from six 1.27 cm diameter columnar muscle samples parallel to the muscle fiber orientation. Columnar samples from each steak were calculated in kg/cm^2^ as the shear force. The method of the AOAC (2005) was used to determine and present the moisture, crude ash, IMF, and protein contents of the LD muscle [31]. A method developed by Wu et al. (2009) was used to measure the fiber diameter and density of muscle fibers [32].

### 2.5. Amino Acid and Fatty Acid Measurements

Approximately 0.2 g of LD and BF muscle from a freeze-dried sample was hydrolyzed in 10 mL of 6 mol/L HCl at 110 °C for 23 h, and shaken for 20 min. The hydrolysate was thinned with deionized water until it reached a total volume of 100 mL. The diluent was diluted 200 times to one milliliter and filtered through a 0.22 m membrane. A Sykam S-433D (from Saikam Scientific Instruments Co., Ltd., Beijing, China) automatic AA analyzer was used to analyze the filtrate. The one-step extraction methylation procedure for LD and BF muscle fatty acid acquisition [4] was performed in a gas chromatograph-mass spectrometer model 7890B-5977B (Agilent, Palo Alto, CA, USA). The column specifications and field parameters were DB-5 (30 m × 0.25 mm × 0.25 µm), and 25 µL (10.337 mg/mL) of C19:0 was used as an internal standard. In this experiment, the microwave oven was programmed to turn on for 5 min at 250 W, move to 630 W for 5 min, switch to 500 W for 20 min, and then turn off for 15 min. The carrier gas for the experiment was helium at a flow rate of 2 mL per minute. A split/splitless injector that operates automatically and has a 1/20 split ratio at a temperature of 270 °C was utilized, with an injection volume of 1 μL. The temperature of the column was set at 70 degrees Celsius for 5 min, and then increased at a rate of 25 degrees Celsius per minute until reaching 200 degrees Celsius, followed by a slower increase of 2 degrees Celsius per minute until reaching 240 degrees Celsius for 10 min. The interface temperature to be controlled at 280 degrees Celsius, while the ion source temperature to be controlled at 230 degrees Celsius; the quadrupole was heated to a temperature of 150 degrees Celsius. An injection volume of 1 microliter was used, and the scan was conducted in full scan mode. The GC/MS data were analyzed using 7890B-5977B Agilent Data software and categorized based on the NIST database.

### 2.6. Statistical Analysis

All date were presented as means and standard errors and tested for normal distribution using the Kolmogorov–Smirnov test on SPSS 22.0. The resulting data were analyzed by one-way ANOVA using the SPSS 22.0 software and one cattle was used as a statistical unit. A value of *p* > 0.05 indicated that the difference was not significant, and *p* < 0.05 indicated that the difference was significant.

## 3. Results

### 3.1. Slaughter Parameter

#### 3.1.1. Carcass Characteristics

Table 2 displays the impact of diet energy on slaughter performance. The LWBS (*p* = 0.03) and carcass weight (*p* = 0.03) significantly differed among the three groups, while the slaughter ratio was identical across the diet energy treatment groups (*p* = 0.33). In the HEG, the 12th rib fat was thicker and the loin muscle area was greater in the 12th rib fat group than in the other two groups (*p* < 0.05). Meat, noncarcass, and the meat-to-bone ratio were unaffected as dietary energy levels increased (*p* > 0.05).

The weights of the majority of the top and medium-top meats showed no variations, with the exception of the highrib and ribeye meats, which were impacted by the energy treatments (*p* < 0.05, Table 3). However, all meat weights increased numerically with increasing dietary energy and were highest in the HEG (*p* > 0.05). The percentage of top-grade cuts in the HEG reached 13.47%, which was significantly greater than that in the MEG (10.62%) and LEG (10.39%, *p* < 0.05).

#### 3.1.2. Organ Indices

Table 4 displays the organ indices. No significant changes in organ indices were observed when dietary energy levels increased (*p* > 0.05).

### 3.2. Meat Quality

Table 5 displays the LD and BF muscle quality traits. While the dietary energy treatments had no effect (*p* > 0.05) on the pH values at 45 min and 24 h, L* and a* values, cooking loss, drip loss, or water-holding capacity, and the b* value in the BF muscle tended to decrease with the energy level of the diet (*p* = 0.07). The day matter, crude ash, and crude protein concentrations of muscle were not influenced by dietary energy treatment (*p* > 0.05). The content of IMF in the LD and BF muscles was dramatically affected by diet energy levels (*p* < 0.05) and was greater in the HEG than in the MEG and LEG, resulting in the lowermost shear force in the LD and BF of the HEG.

### 3.3. Amino Acids and Fatty Acids

#### 3.3.1. Amino Acids

The muscle amino acid profiles were minimally impacted by increasing levels of dietary energy (Table 6 and Table 7). None of the amino acid contents significantly differed in the LD muscle. In the BF muscle, none of the amino acid concentrations were affected by increasing dietary energy levels (*p* > 0.05), except for the histidine (His) concentration in the HEG, which was greater (*p* = 0.06) than that in the MEG and LEG. Furthermore, the muscle contained relatively high levels of leucine (Leu), aspartic acid (Asp), glutamic acid (Glu), proline (Pro), and histidine (His), with Glu being most abundant (*p* > 0.05). Apart from Asp and Pro, the levels of total amino acids (TAAs), essential amino acids (EAAs), and many flavored amino acids (FAAs) generally enhanced as dietary energy levels increased (*p* > 0.05). An interesting finding was that the ratio of umami amino acids (UAAs) to bitter amino acids (BAAs) significantly decreased with increasing levels of dietary energy, while the opposite held true for the ratio of bitter amino acids (BAAs) to umami amino acids (UAAs) (*p* < 0.05). The proportions of amino acids that produced fresh or sweet flavors were more than 67%, and the proportions of amino acids that produced bitter flavors were about 32%. The percentages of essential and nonessential amino acids were roughly 35.7% and 64.3%, respectively. In general, the concentrations of EAAs, FAAs, and TAAs did not show any notable variances between the LD and BF muscles (*p* > 0.05). In all measurements, the BF muscle had greater contents of amino acids, EAAs, and free amino acids, as well as total amino acids, than did the LD muscle (*p* > 0.05).

#### 3.3.2. Fatty Acids

The dietary energy level did little to affect the fatty acid composition of muscle, as shown in Table 8 (*p* > 0.05). In the LD muscle, the majority of fatty acid profiles were not influenced by the high-energy diet. As dietary energy levels increased, there was a decreasing trend in the concentration of C17:1 (*p* = 0.08) and, conversely, an increasing trend in the concentration of C20:4n-6 (*p* = 0.09). Despite having higher levels of monounsaturated fatty acids (MUFAs) and C18:1 cis-9 than the other two groups, HEG had significantly lower levels of saturated fatty acids (SFAs) (*p* < 0.05). In BF muscle, dietary treatments had no effect on saturated fatty acid (SFA) concentrations, monounsaturated fatty acid (MUFA) concentrations, or polyunsaturated fatty acid (PUFA) concentrations (*p* > 0.05). The amounts of saturated fatty acids (SFAs) and unsaturated fatty acids (UFAs) in BF muscle were not significantly affected by increasing energy intake, but the concentrations of SFAs decreased, whereas the concentrations of UFAs tended to increase as dietary energy levels increased (*p* > 0.05).

## 4. Discussion

The performance of slaughtering is a crucial indicator that can efficiently demonstrate the economic profitability of finishing cattle. In evaluating slaughter performance, carcass weight and dressing percentage are the most important factors. Research demonstrated that the energy in an animal’s feed influences its performance at slaughter, with animals tending to gain weight when energy levels rise, leading to heavier carcasses [33,34]. Despite the fact that the LWBS and carcass weight of the HEG were notably greater than those of the other two groups in the present study, there was no difference in the slaughter ratio among the treatments, which was in contrast with previous studies showing that the slaughter ratio improved as the LWBS increased in cattle [19] and lambs [35,36]. These discrepancies might be explained by differences in the breed and growth periods of the selected animals. In addition, a slaughtering rate of 52% in China is set at 0.3 yuan/kg for a gain or loss of one percent higher or lower [37], respectively. In this study, the mean dressing percentage (56.3%) of Honghe yellow cattle was greater than that of Xiangxi yellow cattle (54.2%) [14], yellow breed× Simmental cattle (53.4%) [16], yaks (47.4%) [19], and the threshold. Compared to those in the MEG and LEG, Honghe yellow cattle in the HEG had greater back fat thickness and eye muscle area, suggesting that barn-fed Honghe yellow cattle facilitated carcass fat deposition and resulted in the improvement of carcass traits. Prior studies had shown that an increase in the concentration of dietary energy concentration or energy intake elevated back fat thickness and eye muscle area [19,38], which was in general agreement with our study. A larger eye muscle area means a greater number of primal cuts.

Primal cuts are typically the most valuable part of a carcass, but some markets require specific weights for certain cuts. In this study, the eye and brisket meat weights of Honghe yellow cattle dramatically improved as the dietary energy level rose, while the weights of other meat pieces showed an increasing trend, resulting in a significant increase in the yield of top-grade cuts, which indicated that high levels of dietary energy could help to increase the yield of the main meat. In general, these findings were consistent with previous reports showing that high-energy density diets increased the yield of commercial meat cuts of Xiangxi yellow cattle [15] and yaks [19]. The growth potential of Honghe yellow cattle might be hindered by the energy-deficient conditions under which they graze. According to the current research, the weights of the primal cuts of Honghe yellow cattle increased with increasing dietary energy concentration, suggesting that the energy concentration of the HEG did not reach the maximum capacity of the Honghe yellow cattle, and that the optimum energy requirement for Honghe yellow cattle needs to be further investigated. Studies on the effect of dietary energy levels on the proportion of major parts of Honghe yellow cattle carcasses are scarce, so comparisons cannot be made. Conversely, according to Suarezbelloch et al. (2013) [39], as the dietary NE concentration raised from 2280 kcal/kg to 2420 kcal/kg, the proportion of hams and tenderloins in pigs decreased. These results were probably influenced by the increase in fat deposition during the fattening period as well as the increase in dietary energy density. In this study, Honghe yellow cattle were slaughtered in advance of significant fat deposition, and only muscle development was enhanced.

These results demonstrate a positive impact of ration energy intake on meat quality [19,40,41]. In the current study, when the energy content of the diet increased, the shearing force of Honghe yellow cattle reduced and IMF content enhanced. Similar findings had been reported by other researchers [38,39], who reported that increasing ration energy intake improved meat quality traits in finishing beef cattle. Additionally, a reduction in shear force and an increase in the IMF content were observed after the dietary net energy concentration was increased to 5.32 Nemf from 3.72 Nemf, as reported by Kang et al. (2020) [19]. Nevertheless, in some studies, dietary energy concentration had only few effects on goat and Holstein bull meat quality traits [42,43]. The difference might be attributed to the different levels of energy concentration and growth stages of the animals in the experimental settings. Yu et al. (2022) [3] investigated the meat quality characteristics of grazing Honghe yellow cattle, and the data showed that the average shearing force of grazing Honghe yellow cattle reached 4.05 kg/cm^2^, which is a tough steak [15]. Nevertheless, in this study, the Honghe yellow cattle muscle had an average force of 3.19 kg, which means that the Honghe yellow cattle beef in the HEG reached the intermediate steak standard [15]. It is possible that the decrease in shearing force was due to a higher IMF content. Studies have shown that IMF influences muscle fiber condition, connective tissue composition and content, and muscle proteases, which determine muscle tenderness [44]. As a result of our experiments, we found no differences between the LD and BF muscles in terms of cooking loss, water-holding capacity, or drip loss. Similarly, dietary energy had no effect on cooking loss, water-holding capacity, drip loss, or juiciness [17,45]. However, there is some evidence that meat with a high IMF concentration had decreased drip loss and cooking loss [46]. Those discrepancies in these findings may be caused by differences in age and feed resource.

The amino acid composition of the muscle of Honghe yellow cattle in this study was not affected by dietary energy. The proportion of amino acids producing umami and sweet flavors was greater than 67%, and the proportion of amino acids producing bitter flavors was approximately 32%. The daily requirements of essential and nonessential amino acids for adult males are 0.18 g/kg (EAAs) and 0.48 g/kg (NEAAs), which correspond to EAAs/NEAAs = 37.5% and EAAs/TAAs = 27.3%, respectively [47]. In the present study, the average proportions of EAAs/NEAAs and EAA/TAA in the meat samples were 55.2% and 35.6%, respectively, which were much greater than the recommended ratios of FAO/WHO/UNU [48] and could fully satisfy the needs of adult males; therefore, the beef from Honghe yellow cattle could be an excellent source of protein.

Diet composition impacts the TAAs’ composition of muscle [49]. Research [23] has also shown that rumen microbes convert fatty acids into other compounds, in agreement with the results of the current study. Besides, the fatty acid composition of the LD and BF muscles was reflected in the dietary fatty acids content. The present study revealed that there were unsignificant differences in medium- and long-chain fatty acids among the three treatments, and MUFAs were the predominant fatty acids, followed by SFAs and PUFAs, which was compatible with the findings of other studies [50]. The major SFAs in the LD and BF muscles are palmitic acid C16:0 and C18:0, and the predominant UFAs are C18:1 cis-9 and C18:2n-6. In this study, a decrease in SFAs in the LD muscle and an increase in C18:1 cis-9 were observed with increasing ration energy levels, which was compatible with results from Holstein cattle [16]. This was in accordance with Buchanan et al. (2013) [51]. C18:1 cis-9 increased the IMF ratio and softened fat in the LD muscle [46]. Specifically, the ratio of C22:6n-3 decreased with increasing grain proportion and decreasing forage proportion. This result may be attributed to the fact that the low-energy group contains a high proportion of forage, which includes C18:3n-3, the precursor of n-3 PUFAs, for the production of long-chain n-3 PUFAs [52]. Furthermore, cereal grains are enriched in C18:2n-6, which is the precursor of n-6 PUFAs, for the production of lysergic acid [52]. This result was comparable to that of the study by Wang et al. (2019) [17].

The coefficient of the n-6/n-3 ratio required a minimum score of 7.49, which is far greater than the score recommended by some nutritional advice (<4.0) [53]. The composition of UFAs was influenced by dietary energy levels, according to previous studies [54]. This study revealed that MUFA concentrations increased with the increasing level of dietary energy, which is in agreement with prior research showing that dietary restriction or concentration increases MUFA concentrations in sheep and goats [17,55]. A numerical increase in the UFA–SFA ratio was observed in both LD and BF muscles with increasing dietary energy intake, which might be correlated to corn fat. The UFA–SFA ratio in both the LD and BF muscles was >1.0, indicating that the meat met the modern standards for healthy green foods [53]. The above findings and observations indicated that raising the energy level of the diet altered the fatty acid composition, but did not affect the meat quality or flavor.

## 5. Conclusions

As a result of our study, we found that an increase in ration energy levels improved the LWSB and carcass weight of Honghe yellow cattle, leading to satisfactory slaughter performance. Compared with the LEG, the HEG significantly increased the IMF content of the LD and BF muscles and the eye muscle area but decreased muscle shear force. Dietary energy levels did not affect amino acid composition or muscle fiber diameter. Besides, dietary energy levels significantly reduced the SFAs and C22:6n-3 contents in the LD and BF muscles, but significantly increased the C18:1 cis-9 concentration. Our results provide theoretical guidance for designing feed formulations for yellow cattle, and we recommend an increase in dietary net energy to 5.90 MJ/kg to ensure the economic efficiency of fattening in Honghe yellow cattle. Additionally, we will carry out the molecular mechanism of the effect of different energy levels on the quality of Honghe yellow cattle at a later stage, and will expand the sample size and increase the number of experimental groups to further validate our finding.

## Figures and Tables

**Table 1 foods-13-01316-t001:** As-DM diet composition and nutrient levels.

Item	Treatments		
	LEG	MEG	HEG
Ingredients Composition			
Corn silage	58.0	53.0	49.0
Concentrate supplement	36.0	40.5	43.0
CP powder	2.0	1.9	3.0
Fat powder	4.0	4.6	5.0
Total	100	100	100
Nutrient levels			
Nemf MJ/kg	5.30	5.60	5.90
CP	11.73	11.74	11.72
Ca	0.55	0.58	0.59
P	0.64	0.62	0.63
NDF	46.27	43.64	40.63
ADF	30.8	28.77	26.55

CP = crude protein; ADF = Acid detergent fiber; NDF = neutral detergent fiber; Nemf = comprehensive net energy. Composition of the raw materials for concentrate supplementation were 12% soybean meal, 9% corn protein meal, 5% wine lees, 63% corn, 6% bran, 3% calcium hydrogen phosphate, 1% salt, and 1% premix. The following were used per kilogram of premix: 36 mg Cu, 45 mg Fe, 24 mg Mn, 33 mg Zn, 0.33 mg I, 0.10 mg Co, 0.28 mg Se, 64,000 IU vitamin A, 19,150 IU vitamin D, and 80 mg vitamin E. LEG = low-energy group; MEG = medium-energy group; HEG = high-energy group.

**Table 2 foods-13-01316-t002:** Dietary treatments affect the slaughter performance of Honghe yellow cattle.

Item	LEG	MEG	HEG	SEM	*p*-Value
LWBS, kg	271.83	276	301.33	8.91	0.03
Carcass weight, kg	151.38	154.36	173.35	7.63	0.03
Slaughter rate, %	54.66	56.74	57.55	1.83	0.33
Carcass composition, %					
Meat	76.86	78.05	78.35	1.68	0.45
Noncarcass fat	4.58	4.91	5.57	0.63	0.34
Meat: bone	4.94	5.08	5.19	0.53	0.89
Backfat thickness, mm	6.20	7.01	7.17	0.32	0.05
Loin muscle area, cm^2^	51.03	64.45	64.40	2.88	0.01
Diameter, μm/cm^2^	65.22	67.97	66.30	1.54	0.81

LWBS = live weight before slaughter; LEG = low-energy group (5.30 MJ/kg Nemf); MEG = medium-energy group (5.60 MJ/kg Nemf); HEG = high-energy group (5.90 MJ/kg Nemf).

**Table 3 foods-13-01316-t003:** Dietary treatments affect the commercial cut yield of Honghe yellow cattle.

Item	LEG	MEG	HEG	SEM	*p*-Value
Total meat, kg	124.92	127.01	131.29	4.18	0.36
Top-grade cuts					
Highrib, kg	3.25	3.25	5.09	0.57	0.02
Ribeye, kg	2.44	2.73	2.76	0.24	0.03
Tenderloin, kg	2.81	2.91	3.34	0.28	0.25
Striploin, kg	1.89	1.86	1.94	0.18	0.91
Medium-top-grade cuts					
Brisket, kg	3.70	3.06	4.50	0.62	0.41
Thick flank, kg	3.78	3.85	4.07	0.31	0.65
Shoulder, kg	6.17	6.12	6.76	0.87	0.73
Topside, kg	4.84	4.92	5.54	0.54	0.37
Outside flat, kg	3.96	3.32	4.30	0.47	0.18
Eye round, kg	2.60	2.63	3.18	0.36	0.27
Rump, kg	2.06	2.19	2.16	0.24	0.86
Top-grade cuts yield, %	10.39	10.62	13.47	0.73	0.01
Medium-top-grade cuts, %	26.35	25.93	27.86	2.13	0.66
Total, %	28.65	28.85	31.48	1.68	0.25

LEG = low-energy group (5.30 MJ/kg Nemf); MEG = medium-energy group (5.60 MJ/kg Nemf); HEG = high-energy group (5.90 MJ/kg Nemf).

**Table 4 foods-13-01316-t004:** Dietary treatments affect the organ indices (%) of Honghe yellow cattle.

Item	LEG	MEG	HEG	SEM	*p*-Value
Heart	0.31	0.35	0.35	0.02	0.80
Liver	1.02	1.15	0.98	0.05	0.35
Spleen	0.24	0.30	0.29	0.02	0.42
Lungs	0.97	0.96	0.80	0.06	0.49
Kidney	0.15	0.16	0.16	0.00	0.69
Rumen	1.80	1.65	1.56	0.10	0.68
Reticulum	0.23	0.21	0.22	0.01	0.82
Omasum	0.68	0.63	0.60	0.04	0.72
Abomasum	0.26	0.34	0.30	0.03	0.69
Total stomach	3.01	2.80	2.68	0.16	0.73
Small intestine	0.18	0.19	0.15	0.01	0.25
Large intestine	0.17	0.16	0.13	0.01	0.39

LEG = low-energy group (5.30 MJ/kg Nemf); MEG = medium-energy group (5.60 MJ/kg Nemf); HEG = high-energy group (5.90 MJ/kg Nemf).

**Table 5 foods-13-01316-t005:** Dietary treatments affect the meat quality of Honghe yellow cattle.

Item	LD					BF				
	LEG	MEG	HEG	SEM	*p*-Value	LEG	MEG	HEG	SEM	*p*-Value
pH 45 min	6.63	6.53	6.59	0.05	0.76	6.66	6.60	6.37	0.08	0.38
pH 24 h	5.66	5.72	5.63	0.02	0.24	5.64	5.63	5.61	0.01	0.62
Cooking loss, %	28.95	28.59	27.07	0.56	0.39	29.16	28.37	28.46	0.47	0.81
Drip loss, %	5.27	5.09	5.00	0.09	0.49	5.07	5.01	4.98	0.03	0.58
Water-holding capacity	45.32	45.47	43.12	0.55	0.14	46.99	46.66	46.97	0.84	0.99
Shear force, kg	3.67	3.61	3.16	0.09	0.01	4.05	3.64	3.19	0.16	0.04
Color parameters										
Lightness, L*	34.72	36.09	36.46	0.54	0.43	38.65	35.95	38.88	0.81	0.30
Redness, a*	17.42	16.98	16.32	0.42	0.63	15.86	16.10	16.06	0.25	0.94
Yellowness, b*	5.96	5.46	5.52	0.14	0.32	5.90	5.88	5.50	0.09	0.07
Nutritional characteristics										
Dry matter, %	29.17	30.23	32.33	0.73	0.22	28.34	29.92	31.97	0.88	0.26
Crude ash, %	1.00	0.98	1.21	0.06	0.34	1.00	1.00	1.03	0.02	0.86
Protein, %	85.02	87.66	86.76	0.94	0.57	75.57	74.81	75.07	1.71	0.99
Intramuscular fat, %	16.02	18.07	21.90	1.03	0.03	8.01	9.46	15.17	1.19	0.01

LD = longissimus dorsi muscle; BF = biceps femoris; LEG = low-energy group (5.30 MJ/kg Nemf); MEG = medium-energy group (5.60 MJ/kg Nemf); HEG = high-energy group (5.90 MJ/kg Nemf).

**Table 6 foods-13-01316-t006:** Dietary treatments affect the amino acid composition in the muscles of Honghe yellow cattle (g/100 g).

Item	LD					BF				
	LEG	MEG	HEG	SEM	*p*-Value	LEG	MEG	HEG	SEM	*p*-Value
Essential										
Lysine	1.05	1.04	1.13	0.03	0.39	1.90	2.04	2.11	0.05	0.30
Valine	1.24	1.29	1.36	0.02	0.15	1.32	1.40	1.44	0.03	0.39
Histidine	1.82	1.91	1.95	0.03	0.30	0.99	1.10	1.20	0.04	0.06
Leucine	2.04	2.13	2.18	0.04	0.37	2.13	2.29	2.38	0.06	0.26
Isoleucine	1.22	1.29	1.35	0.03	0.12	1.31	1.40	1.43	0.03	0.47
Methionine	1.23	1.29	1.36	0.03	0.13	0.16	0.19	0.19	0.01	0.67
Phenylalanine	1.01	1.07	1.10	0.02	0.22	1.07	1.14	1.18	0.03	0.34
Threonine	1.16	1.22	1.25	0.02	0.32	1.22	1.31	1.36	0.04	0.32
Nonessential										
Aspartic acid	2.33	2.43	2.50	0.04	0.33	2.72	2.62	2.43	0.07	0.27
Glutamic acid	4.08	4.33	4.40	0.08	0.26	4.28	4.6	4.78	0.13	0.31
Cysteine	0.21	0.22	0.21	0.00	0.30	0.21	0.23	0.23	0.01	0.32
Alanine	1.44	1.50	1.55	0.03	0.24	1.53	1.62	1.67	0.04	0.36
Glycine	1.08	1.09	1.18	0.03	0.24	1.22	1.19	1.20	0.02	0.83
Serine	0.99	1.03	1.03	0.02	0.56	1.01	1.09	1.14	0.03	0.22
Proline	2.33	2.35	2.40	0.03	0.71	2.13	2.24	2.14	0.03	0.19
Arginine	1.53	1.60	1.66	0.03	0.23	1.62	1.73	1.78	0.04	0.34
Tyrosine	0.85	0.90	0.86	0.02	0.59	0.86	0.96	0.99	0.03	0.21

LD = longissimus dorsi muscle; BF = biceps femoris; LEG = low-energy group (5.30 MJ/kg Nemf); MEG = medium-energy group (5.60 MJ/kg Nemf); HEG = high-energy group (5.90 MJ/kg Nemf).

**Table 7 foods-13-01316-t007:** Dietary treatments affect the flavor amino acid composition in the muscles of Honghe yellow cattle (g/100 g).

Item	LD					BF				
	LEG	MEG	HEG	SEM	*p*-Value	LEG	MEG	HEG	SEM	*p*-Value
Total AAs	24.56	25.56	26.28	0.42	0.26	25.67	27.17	27.66	0.59	0.42
EAAs	8.67	9.07	9.36	0.17	0.26	9.10	9.77	10.09	0.27	0.34
NEAAs	15.89	16.49	16.92	0.25	0.27	16.57	17.40	17.57	0.33	0.49
EAAs/TAAs	35.28	35.46	35.60	0.10	0.46	35.43	35.96	36.46	0.23	0.19
EAAs/NEAAs	54.52	54.94	55.29	0.23	0.46	54.88	56.14	57.39	0.55	0.19
Umami AAs	6.42	6.76	6.90	0.12	0.28	7.00	7.25	7.20	0.16	0.83
Sweet AAs	12.91	13.42	13.77	0.20	0.21	13.29	14.12	14.40	0.29	0.31
Bitter AAs	9.10	9.48	9.80	0.18	0.30	9.45	10.20	10.59	0.28	0.27
UAAs/TAAs	26.13	26.42	26.25	0.07	0.31	27.24	26.69	26.03	0.22	0.04
SAAs/TAAs	52.57	52.54	52.39	0.11	0.83	51.78	51.96	52.10	0.13	0.69
BAAs/TAAs	37.07	37.08	37.29	0.09	0.66	36.81	37.56	38.27	0.26	0.04

LD = longissimus dorsi muscle; BF = biceps femoris; LEG = low-energy group (5.30 MJ/kg Nemf); MEG = medium-energy group (5.60 MJ/kg Nemf); HEG = high-energy group (5.90 MJ/kg Nemf); EAAs = essential amino acids; NEAAs = nonessential amino acids; Umami AAs include Glutamic acid, Aspartic acid; Sweet AAs include Glycine, Alanine, Serine, Threonine, Proline; Bitter AAs include Arginine, Histidine, Leucine, Isoleucine, Methionine, Phenylalanine, Tyrosine, Valine.

**Table 8 foods-13-01316-t008:** Dietary treatments affect the percentages of the main fatty acids in the muscles of Honghe yellow cattle (%).

Item	LD					BF				
	LEG	MEG	HEG	SEM	*p*-Value	LEG	MEG	HEG	SEM	*p*-Value
C14:0	1.96	1.86	1.89	0.15	0.97	1.80	1.75	2.06	0.16	0.74
C15:0	0.24	0.26	0.26	0.01	0.82	0.30	0.29	0.31	0.01	0.88
C16:0	27.56	27.53	24.73	0.83	0.31	25.81	26.91	24.30	0.91	0.56
C17:0	0.98	0.93	0.70	0.09	0.46	0.98	0.93	0.70	0.09	0.46
C18:0	16.53	14.20	14.41	0.57	0.19	16.22	14.56	15.37	0.65	0.65
C20:0	0.47	0.40	0.36	0.06	0.80	0.47	0.50	0.46	0.04	0.94
C21:0	0.58	0.50	0.50	0.03	0.58	0.45	0.41	0.38	0.02	0.52
C22:0	0.37	0.32	0.30	0.02	0.35	0.43	0.41	0.40	0.02	0.88
C14:1	1.00	1.05	1.20	0.06	0.40	1.67	1.59	1.73	0.07	0.75
C15:1	0.15	0.12	0.12	0.01	0.27	0.19	0.16	0.16	0.01	0.29
C16:1	2.99	3.24	3.51	0.17	0.51	3.99	4.24	4.51	0.17	0.51
C17:1	0.72	0.63	0.54	0.04	0.08	0.56	0.61	0.64	0.03	0.66
C18:1 cis-9	40.01	42.23	44.54	0.78	0.03	41.06	41.42	42.49	0.84	0.82
C18:1 trans	2.37	2.03	1.93	0.13	0.45	2.03	2.14	2.09	0.08	0.89
C22:1	0.26	0.33	0.37	0.03	0.31	0.46	0.42	0.50	0.02	0.49
C18: 2n-6 trans	0.15	0.14	0.20	0.01	0.07	0.18	0.16	0.17	0.01	0.80
C18: 2n-6	2.73	3.38	3.44	0.18	0.23	2.46	2.46	2.73	0.11	0.58
C18: 3n-3	0.22	0.30	0.25	0.02	0.26	0.12	0.16	0.12	0.01	0.41
C18: 3n-6	0.05	0.05	0.05	0.00	0.83	0.06	0.07	0.07	0.00	0.52
C20:3n-6	0.08	0.09	0.10	0.01	0.47	0.07	0.07	0.09	0.01	0.57
C20:3n-3	0.02	0.02	0.03	0.00	0.21	0.05	0.06	0.06	0.00	0.34
C20:4n-6	0.40	0.32	0.48	0.03	0.09	0.40	0.46	0.46	0.02	0.33
C20:5n-3	0.05	0.05	0.04	0.01	0.61	0.06	0.08	0.07	0.00	0.15
C22:6n-3	0.14	0.11	0.09	0.01	0.01	0.16	0.14	0.14	0.01	0.57
ΣSFAs	48.71	46.00	43.15	1.00	0.04	46.46	45.75	43.99	1.13	0.71
ΣUFAs	51.29	54.01	56.85	0.99	0.45	53.53	54.25	56.01	1.13	0.72
ΣMUFAs	47.51	49.64	52.21	0.77	0.01	49.98	50.58	52.11	0.89	0.66
ΣPUFAs	3.85	4.45	4.68	0.19	0.20	3.56	3.67	3.90	0.11	0.53
UFAs/SFAs	1.05	1.18	1.32	0.14	0.48	1.16	1.19	1.29	0.06	0.70
PUFAs/MUFAs	0.08	0.09	0.09	0.00	0.44	0.07	0.07	0.07	0.01	0.89
n-6 PUFAs	3.41	3.97	4.27	0.20	0.22	3.17	3.23	3.51	0.11	0.48
n-3 PUFAs	0.43	0.49	0.41	0.02	0.12	0.39	0.44	0.38	0.02	0.59
n-6/n-3 PUFAs	8.01	8.18	10.70	0.71	0.24	8.27	7.49	9.36	0.56	0.45

LD = longissimus dorsi muscle; BF = biceps femoris; LEG = low-energy group (5.30 MJ/kg Nemf); MEG = medium-energy group (5.60 MJ/kg Nemf); HEG = high-energy group (5.90 MJ/kg Nemf); ΣSFAs = saturated fatty acids (C14:0, C15:0, C16:0, C17:0, C18:0, C20:0, C21:0, C22:0); ΣUFAs = unsaturated fatty acids (C14:1, C15:1, C16:1, C17:1, C18:1 cis-9, C18:1 trans, C22:1, C18:2n-6trans, C18:2n-6, C18:3n-3, C18:3n-6, C20:3n-3, C20:4n-6, C20:5n-3, C22:6n-3); ΣMUFAs = monounsaturated fatty acids (C14:1, C15:1, C16:1, C17:1, C18:1 cis-9, C18:1 trans, C22:1); ΣPUFAs = polyunsaturated fatty acids (C18:2n-6trans, C18:2n-6, C18:3n-3, C18:3n-6, C20:3n-3, C20:4n-6, C20:5n-3, C22:6n-3); n-6 PUFAs = C18:2n-6trans + C18:2n-6 + C18:3n-6 + C20:4n-6; n-3 PUFAs = C18:3n-3 + C20:3n-3 + C20:5n-3 + C22:6n-3.

## Data Availability

The original contributions presented in the study are included in the article, further inquiries can be directed to the corresponding author.

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
