# Peer review of "Impact of Various Ration Energy Levels on the Slaughtering Performance, Carcass Characteristics, and Meat Qualities of Honghe Yellow Cattle"

_foods, 2024, doi:10.3390/foods13091316_

Round 1
Reviewer 1 Report
Comments and Suggestions for Authors
The manuscript entitled “Comparisons of the slaughter performance, carcass traits, and meat quality of Honghe yellow cattle consuming different amounts of energy” presents interesting issue, however some corrections are needed.
– Lines 36-37 – “HongheAs the demand for beef increases in China, beef production is insufficient and beef and beef products are increasingly imported from other countries such as Brazil, New Zealand and Argentina to meet the beef demand” – some sentences require attention.
– Line 65 – “IMF content” – All abbreviations when used for the first time must be explained
– Lines 166-167 –“ and then the results are expressed 166 as "mean ± standard deviation"” Was the normality of distribution tested? The information about it should be added and authors should be consequent. If data have normal distribution, they should be treated as such, if not, nonparametric tests should be applied. Please specify it.
– Lines 181-183 “LEG= high energy group (5.30 MJ/kg Nemf); MEG= medium energy group (5.60 MJ/kg Nemf); HEG= low energy group” Are you sure about this? Because the abbreviations look different (L – for low; H for high). Could you please check all the signatures under the tables?
– The entire article requires some editorial attention
– The novelty and significance of this research need to be emphasized.
Author Response
|
Response to Reviewer 1 Comments
|
||
|
1. Summary |
|
|
|
Thank you very much for taking the time to review this manuscript. Please find the detailed responses below and the corresponding revisions/corrections highlighted/in track changes in the re-submitted files.
|
||
|
2. Questions for General Evaluation |
Reviewer’s Evaluation |
Response and Revisions |
|
Does the introduction provide sufficient background and include all relevant references? |
Can be improved |
Improvements made |
|
Are all the cited references relevant to the research? |
Can be improved |
Improvements made |
|
Is the research design appropriate? |
Can be improved |
Improvements made |
|
Are the methods adequately described? |
Can be improved |
Improvements made |
|
Are the results clearly presented? |
Can be improved |
Improvements made |
|
Are the conclusions supported by the results? |
Can be improved |
Improvements made |
|
3. Point-by-point response to Comments and Suggestions for Authors |
||
|
Comments 1: Lines 36-37 – “HongheAs the demand for beef increases in China, beef production is insufficient and beef and beef products are increasingly imported from other countries such as Brazil, New Zealand and Argentina to meet the beef demand” – some sentences require attention. |
||
|
Response 1: Thank you for pointing this out. I agree with this comment. By your review comments, we found that the first sentence in the introduction did have the problem you described, and at the same time, in order to better emphasize the effects of different energy levels on Honghe yellow cattle, we teamed up the introduction part to make additions. In addition, we have scrutinized and corrected the whole text to avoid similar situations from recurring. The changes are as follows: As increased demand for beef in China is met by insufficient beef production, beef and beef products are increasingly imported from other countries, such as Brazil, New Zealand and Argentina, to meet beef demand [1]. In China, there is a rich resource of cattle breeds, the largest number of cattle breeds in the world, including 53 local breeds of cattle, 7 breeds of cattle bred by China, and 13 imported breeds of cattle [2]. Mention exactly where in the revised manuscript this change can be found – lines 36-40. |
||
|
Comments 2: Line 65 – “IMF content” – All abbreviations when used for the first time must be explained. |
||
|
Response 2: Thank you for pointing this out. I agree with this comment. By your review comments, Abbreviations should be explained the first time used, which is a sign of academic norms and rigor, and we have corrected for this error, while similar issues throughout the text have been checked and corrected. Mention exactly where in the revised manuscript this change can be found – lines 59. |
||
|
Comments 3: Lines 166-167 -“ and then the results are expressed 166 as "mean ± standard deviation"” Was the normality of distribution tested? The information about it should be added and authors should be consequent. If data have normal distribution, they should be treated as such, if not, nonparametric tests should be applied. Please specify it. |
||
|
Response 3: Agree. Thank you for your guidance and suggestions. We have used SPSS 22.0 software to perform one way ANOVA on the general linear model and the data obtained are expressed as mean and standard error (SEM) and tested for normal distribution using Kolmogorov-Smirnov test of SPSS 22.0 and also in among the materials and methods has been modified and we have applied the data processing mentioned in the materials and methods and the results are in accordance with the normality distribution model. Mention exactly where in the revised manuscript this change can be found – lines 187-191. |
||
|
Comments 4: Lines 181-183 “LEG= high energy group (5.30 MJ/kg Nemf); MEG= medium energy group (5.60 MJ/kg Nemf); HEG= low energy group” Are you sure about this? Because the abbreviations look different (L – for low; H for high). Could you please check all the signatures under the tables? |
||
|
Response 4: Agree. Thank you all so much for the note of caution. It was indeed an oversight on my part to make this error, and we have corrected the issues you pointed out and corrected the tables throughout the text as well. Mention exactly where in the revised manuscript this change can be found – lines 118-119, lines 203-204, lines 214-215, lines 221-222, lines 233-235, lines 261-2263, lines 266-267, lines 288-289, lines 293-294. |
||
|
4. Response to Comments on the Quality of English Language |
||
|
Point 1: The entire article requires some editorial attention. |
||
|
Response 1: Agreed. As per your suggestion, we have checked and corrected the whole text in detail. In addition, this manuscript has had the full text checked in detail by a professional teacher and corrected for statements and grammatical problems. |
||
|
5. Additional clarifications |
||
|
Point 1: The novelty and significance of this research need to be emphasized. |
||
|
Response 1: Thank you for pointing this out. We agree with your comments. Through your comments, we re-read the manuscript carefully and reviewed a large amount of related literature, and we believe that the novelty of this study is that, although, more studies have been carried out on other cattle breeds in China, and the experimental results have obtained the appropriate level of energy addition, which is of guiding significance for the production of these breeds of cattle. However, the results of these studies are not applicable to the production practice of Honghe yellow cattle due to the different dietary energy requirements of different breeds and different growth stages of cattle. In addition, there are few studies on the effects of dietary energy levels on Honghe yellow cattle. Therefore, we carried out a study on the effects of different dietary energy levels on slaughter performance, carcass traits and meat quality of Honghe yellow cattle, using Honghe yellow cattle as test animals, which not only provided specific guidance on the improvement of beef quality of Honghe yellow cattle, but also provided a scientific basis for the sustainable development of the beef cattle industry. Mention exactly where in the revised manuscript this change can be found – lines 78-86. |
||
Reviewer 2 Report
Comments and Suggestions for Authors
The paper is interesting, but you used only five animals/samples for each treatment. In my opinion it cannot be considered representative. Normally they are required from 10 to 12 animals for each treatment.
Author Response
For research article
|
Response to Reviewer 2 Comments
|
||
|
1. Summary |
|
|
|
Thank you very much for taking the time to review this manuscript. Please find the detailed responses below and the corresponding revisions/corrections highlighted/in track changes in the re-submitted files.
|
||
|
2. Questions for General Evaluation |
Reviewer’s Evaluation |
Response and Revisions |
|
Does the introduction provide sufficient background and include all relevant references? |
Can be improved |
Improvements made |
|
Are all the cited references relevant to the research? |
Can be improved |
Improvements made |
|
Is the research design appropriate? |
Can be improved |
Improvements made |
|
Are the methods adequately described? |
Can be improved |
Improvements made |
|
Are the results clearly presented? |
Can be improved |
Improvements made |
|
Are the conclusions supported by the results? |
Can be improved |
Improvements made |
|
3. Point-by-point response to Comments and Suggestions for Authors |
||
|
Comments 1: The paper is interesting, but you used only five animals/samples for each treatment. In my opinion it cannot be considered representative. Normally they are required from 10 to 12 animals for each treatment. |
||
|
Response 1: Thank you for pointing this out. We agree with your comments. First, we recognize that ideally the sample size should be larger for each treatment group to enhance the statistical efficacy of the study. However, in this study, the sample size was limited by the following aspects: 1. scarcity of special breeds: the Honghe yellow cattle is a special local breed with a limited number of healthy adult individuals available for the study, especially under the experimental requirements. This directly limits the number of samples we can include in the study. 2. Preliminary exploratory study: The purpose of this study was to conduct a preliminary exploration aimed at evaluating the effects of different energy levels on slaughter performance, carcass traits and meat quality of Honghe yellow cattle. We expect that this study will provide the basis and direction for subsequent larger-scale studies. 3. Despite the small sample size, we took the following measures to ensure the reliability and validity of the study results: Consistency of results: Despite the limited sample size, the trends and patterns we observed were very consistent across all individuals, which enhanced the credibility of our findings. Literature support: our findings are consistent with the published results of related studies, which further justifies the plausibility and credibility of our results. Therefore, we consider the number of animals in this study to be biologically significant and the data to be highly credible. The references are as follows: 1.Huang Q, Wang S, Yang X, Han X, Liu Y, Khan NA, Tan Z. Effects of organic and inorganic selenium on selenium bioavailability, growth performance, antioxidant status and meat quality of a local beef cattle in China. Front Vet Sci. 2023 Apr 27;10:1171751. doi: 10.3389/fvets.2023.1171751. 2.Lin H, Zhao S, Han X, Guan W, Liu B, Chen A, Sun Y, Wang J. Effect of static magnetic field extended supercooling preservation on beef quality. Food Chem. 2022 Feb 15;370:131264. doi: 10.1016/j.foodchem.2021.131264. 3.Ueda S, Hosoda M, Kasamatsu K, Horiuchi M, Nakabayashi R, Kang B, Shinohara M, Nakanishi H, Ohto-Nakanishi T, Yamanoue M, Shirai Y. Production of Hydroxy Fatty Acids, Precursors of γ-Hexalactone, Contributes to the Characteristic Sweet Aroma of Beef. Metabolites. 2022 Apr 6;12(4):332. doi: 10.3390/metabo12040332. 4.Bulkaini B, Dahlanuddin D, Ariana T, Kisworo D, Maskur M, Mastur M. Marbling score, cholesterol, and physical-chemical content of male Bali beef fed fermented pineapple peel. J Adv Vet Anim Res. 2022 Sep 30;9(3):419-431. doi: 10.5455/javar.2022.i610. 5.Chanjula P, Wungsintaweekul J, Chiarawipa R, Phesatcha K, Suntara C, Prachumchai R, Pakdeechanuan P, Cherdthong A. Effects of Supplementing Finishing Goats with Mitragyna speciosa (Korth) Havil Leaves Powder on Growth Performance, Hematological Parameters, Carcass Composition, and Meat Quality. Animals (Basel). 2022 Jun 26;12(13):1637. doi: 10.3390/ani12131637. 6.Li L, Zhu Y, Wang X, He Y, Cao B. Effects of different dietary energy and protein levels and sex on growth performance, carcass characteristics and meat quality of F1 Angus × Chinese Xiangxi yellow cattle. J Anim Sci Biotechnol. 2014 Apr 16;5(1):21. doi: 10.1186/2049-1891-5-21. 7.Yu Y, Wang S, Lu Q, Tao Y, Et Al. Comparison Of Slaughter Performances And Meat Qualities Of Honghe Yellow Cattle At Different Ages. Revista Brasileira De Zootecnia. 2022, 51. |
||
Reviewer 3 Report
Comments and Suggestions for Authors
This study evaluated the effects of different dietary energy levels on the slaughter performance, carcass characteristics, and meat quality of Honghe yellow cattle.
1) The first of all, the title of the manuscript should be rephrased. E.g., Effects of different dietary energy levels…
2) Lines 36-38: the sentence is unclear and should be rephrased.
3) Line 60: “dietary nutrition” please change it.
4) In the Introduction, the authors listed previous studies regarding the effects of diet on many cattle breeds. However, previous relevant research data for Honghe yellow cattle is missing. In this context, what is the novelty of this study?
5) Lines 82-83: please add the place where and year when the study was conducted.
6) The composition and nutrient level of experimental diets must be provided in the manuscript, not in the supplementary material.
7) Within the statistical section, authors listed that results are presented as mean and standard deviations. However, the results are expressed as SEM in the Tables. Also, data on the number of samples used for analysis is missing.
8) Lines 172-173, 209-211: no need to repeat data presented in the Table in the manuscript's text.
9) Please add more data on what organ indices mean. Weights?
10) Line 216: Add “profiles” or change to “Nutritional characteristics”
11) Consider presenting Tables with non-significant results within the Supplementary material.
12) Add units for results expressed within Table 5 and Table 6.
13) Line 321: what does “dietary energy concentration” mean?
Comments on the Quality of English LanguageSome corrections are needed.
Author Response
For research article
|
Response to Reviewer 3 Comments
|
||
|
1. Summary |
|
|
|
Thank you very much for taking the time to review this manuscript. Please find the detailed responses below and the corresponding revisions/corrections highlighted/in track changes in the re-submitted files.
|
||
|
2. Questions for General Evaluation |
Reviewer’s Evaluation |
Response and Revisions |
|
Does the introduction provide sufficient background and include all relevant references? |
Can be improved |
Improvements made |
|
Are all the cited references relevant to the research? |
Can be improved |
Improvements made |
|
Is the research design appropriate? |
Can be improved |
Improvements made |
|
Are the methods adequately described? |
Can be improved |
Improvements made |
|
Are the results clearly presented? |
Can be improved |
Improvements made |
|
Are the conclusions supported by the results? |
Can be improved |
Improvements made |
|
3. Point-by-point response to Comments and Suggestions for Authors |
||
|
Comments 1: The first of all, the title of the manuscript should be rephrased. E.g., Effects of different dietary energy levels… |
||
|
Response 1: Thank you for pointing this out. I agree with this comment. We have reviewed the full text of the study and consulted some of the relevant literature, and the title of this research manuscript really does not emphasize the main idea of the article. Therefore, we have revised the title of the article as follows: Impact of different ration energy levels on the slaughtering performance, carcass characteristics and meat qualities of Honghe yellow cattle. Mention exactly where in the revised manuscript this change can be found - lines 2-4. |
||
|
Comments 2: Lines 36-38: the sentence is unclear and should be rephrased. |
||
|
Response 2: Thank you for pointing this out. I agree with this comment. By your review comments, we found that the first sentence in the introduction did have the problem you described, and at the same time, in order to better emphasize the effects of different energy levels on Honghe yellow cattle, we teamed up the introduction part to make additions. In addition, we have scrutinized and corrected the whole text to avoid similar situations from recurring. The changes are as follows: As increased demand for beef in China is met by insufficient beef production, beef and beef products are increasingly imported from other countries, such as Brazil, New Zealand and Argentina, to meet beef demand [1]. In China, there is a rich resource of cattle breeds, the largest number of cattle breeds in the world, including 53 local breeds of cattle, 7 breeds of cattle bred by China, and 13 imported breeds of cattle [2]. Mention exactly where in the revised manuscript this change can be found – lines 36-40. |
||
|
Comments 3: Line 60: “dietary nutrition” please change it. |
||
|
Response 3: Agree. According to your suggestion, we have sorted out the content of the full text of the study and reviewed some related literature and found that the dietary nutrition could not emphasize the theme of this study, therefore, We have revised this phrase to Concentration of ration energy plays an important role in slaughter performance, carcass qualities and meat quality traits. Mention exactly where in the revised manuscript this change can be found – lines 63-64. |
||
|
Comments 4: In the Introduction, the authors listed previous studies regarding the effects of diet on many cattle breeds. However, previous relevant research data for Honghe yellow cattle is missing. In this context, what is the novelty of this study? |
||
|
Response 4: Thank you for pointing this out. We agree with this comment. Through your comments, we re-read the manuscript carefully and reviewed a lot of related literature, and we think the novelty of this study is as follows: although, more studies were conducted on other cattle breeds in China, and the experimental results also obtained the appropriate energy addition level and have guiding significance for the production of these breeds. However, these results could not be applied to the Honghe yellow cattle. Therefore, we carried out a study on the effects of different dietary energy levels on the slaughter performance, carcass traits and meat quality of the Honghe yellow cattle with the Honghe yellow cattle as the test animal, with the purpose of better guiding the actual production of the Honghe yellow cattle, and thus improving the production efficiency. Mention exactly where in the revised manuscript this change can be found – lines 77-86. |
||
|
Comments 5: Lines 82-83: please add the place where and year when the study was conducted. |
||
|
Response 5: Thank you for pointing this out. I agree with this comment. We have added and clarified the Materials and Methods section of the article, and we believe that revisions based on your comments will make the article more complete. Mention exactly where in the revised manuscript this change can be found – lines 91-93. |
||
|
Comments 6: The composition and nutrient level of experimental diets must be provided in the manuscript, not in the supplementary material. |
||
|
Response 6: Thank you for pointing this out. I agree with this comment. We uploaded the diet formulas to the attached table based on the 2022 article published in foods "Meat Quality and Muscle Tissue Proteome of Crossbred Bulls Finished under Feedlot Using Wet Distiller Grains By-Product" and then I checked recent articles in Foods and other journals and I think your suggestion is correct. As a result, we have added the diet formulation table for this trial to Table 1 of the manuscript. Mention exactly where in the revised manuscript this change can be found – lines 112-118. |
||
|
Comments 7: Within the statistical section, authors listed that results are presented as mean and standard deviations. However, the results are expressed as SEM in the Tables. Also, data on the number of samples used for analysis is missing. |
||
|
Response 7: Agree. Thank you for your guidance and suggestions. By your review comments, we have described the number of samples analyzed in this experiment in the 2.3 Slaughter parameter measurements section of Materials and Methods in the manuscript. Additionally, we have used SPSS 22.0 software to perform one way ANOVA on the general linear model and the data obtained are expressed as mean and standard error (SEM) also in among the materials and methods has been modified. Mention exactly where in the revised manuscript this change can be found – lines 187-189. |
||
|
Comments 8: Lines 172-173, 209-211: no need to repeat data presented in the Table in the manuscript's text. |
||
|
Response 8: Thank you for pointing this out. I agree with this comment. Initially, the focus of the article was described in duplicate in order to highlight the key data from this trial and to give the reader a better understanding of the study. After reading your suggestion, we believe that removing the data shown in the table in the duplicate manuscript text would make the article more concise and clearer. Therefore, based on your suggestion, we have removed the data shown in the table in the duplicate manuscript text and have checked the full text to prevent similar situations. Mention exactly where in the revised manuscript this change can be found – lines 195-196, and lines 2227-230. |
||
|
Comments 9: Please add more data on what organ indices mean. Weights? |
||
|
Response 9: Thank you for pointing this out. Organ weight was not chosen for this experiment, but rather organ index was chosen mainly because organ index can provide a standardized method of comparison, making organ size comparisons between animals of different body sizes or body weights more comparable, and can reveal the trends and characteristics of animal growth and development more intuitively. In contrast, direct comparison of organ weights may be influenced by the body size and weight of the animal and may lead to inaccurate conclusions. This has been demonstrated in other literature. The relevant literature is as follows: 1.Wang Q, Wang Y, Hussain T, Dai C, Li J, Huang P, Li Y, Ding X, Huang J, Ji F, Zhou H, Yang H. Effects of dietary energy level on growth performance, blood parameters and meat quality in fattening male Hu lambs. J Anim Physiol Anim Nutr (Berl). 2020 Mar;104(2):418-430. doi: 10.1111/jpn.13278. |
||
|
Comments 10: Line 216: Add “profiles” or change to “Nutritional characteristics”. |
||
|
Response 10: Agree. I have, accordingly, revised this point. Mention exactly where in the revised manuscript this change can be found – lines 231. |
||
|
Comments 11: Consider presenting Tables with non-significant results within the Supplementary material. |
||
|
Response 11: Thank you for pointing this out. I agree with this comment. However, the tables presented in this study are all significant indicators and all data have been put into the article, including both significant and non-significant results. |
||
|
Comments 12: Add units for results expressed within Table 5 and Table 6. |
||
|
Response 12: Agree. I have, accordingly, revised this point. Based on your suggestion, we double-checked all the tables in the manuscript, and I also found the same problem with Table 8 and Table 9, which we have revised. Mention exactly where in the revised manuscript this change can be found – line 259, line 264, line 286 and line 291. |
||
|
Comments 13: Line 321: what does “dietary energy concentration” mean? |
||
|
Response 13: Thank you for pointing this out. The "dietary energy concentration" in the article mainly explains the reason why the results of this experiment are opposite to those of other studies. The results of our experiment showed that with the increase of dietary energy concentration, some of the high-quality meat pieces of Honghe yellow cattle increased significantly, while the opposite result was observed in fattening pigs. This is mainly due to two reasons, one is the increase of fat deposition capacity in fattening pigs, and the other is the increase of fat deposition capacity with the increase of dietary concentration, these reasons lead to the fat deposition capacity but reduce the yield of quality meat pieces. Therefore, "dietary energy concentration" in the article is the energy level of the diet. Mention exactly where in the revised manuscript this change can be found - lines 319-321. |
||
|
4. Response to Comments on the Quality of English Language |
||
|
Point 1: The entire article requires some editorial attention. |
||
|
Response 1: Agreed. As per your suggestion, we have checked and corrected the whole text in detail. In addition, this manuscript has had the full text checked in detail by a professional teacher and corrected for statements and grammatical problems. |
||
|
5. Additional clarifications |
||
|
Point 1: The novelty and significance of this research need to be emphasized. |
||
|
Response 1: Thank you for pointing this out. We agree with this comment. Through your comments, we re-read the manuscript carefully and reviewed a lot of related literature, and we think the novelty of this study is as follows: although, more studies were conducted on other cattle breeds in China, and the experimental results also obtained the appropriate energy addition level and have guiding significance for the production of these breeds. However, the results of these studies are not applicable to the production practices of Honghe yellow cattle because of the different dietary energy requirements of different breeds and growth stages of cattle. Additionally, there are few studies on the effects of dietary energy levels on Honghe yellow cattle. Therefore, we carried out a study on the effects of different dietary energy levels on the slaughter performance, carcass traits and meat quality of the Honghe yellow cattle with the Honghe yellow cattle as the test animal, with the purpose of better guiding the actual production of the Honghe yellow cattle, and thus improving the production efficiency. Mention exactly where in the revised manuscript this change can be found – lines 78-86. |
||
Round 2
Reviewer 1 Report
Comments and Suggestions for Authors
All my comments have been taken into account. I have no further questions.
Author Response
|
Response to Reviewer 1 Comments
|
||
|
1. Summary |
|
|
|
Thank you very much for taking the time to review this manuscript. Please find the detailed responses below and the corresponding revisions/corrections highlighted/in track changes in the re-submitted files.
|
||
|
2. Questions for General Evaluation |
Reviewer’s Evaluation |
Response and Revisions |
|
Does the introduction provide sufficient background and include all relevant references? |
Yes |
Improvements made |
|
Are all the cited references relevant to the research? |
Yes |
Improvements made |
|
Is the research design appropriate? |
Yes |
Improvements made |
|
Are the methods adequately described? |
Yes |
Improvements made |
|
Are the results clearly presented? |
Yes |
Improvements made |
|
Are the conclusions supported by the results? |
Yes |
Improvements made |
|
3. Point-by-point response to Comments and Suggestions for Authors |
||
|
Comments 1: All my comments have been taken into account. I have no further questions. |
||
|
Response 1: Thank you for your guidance and dedication. At the same time, we are honored to have your approval of the research content of this trial. |
||
Reviewer 2 Report
Comments and Suggestions for Authors
To be such a statistical comparison must be based on enough replicas. In this sector they are normally used from 10-12 replicas. Up to 8 cannot be accepted no less.
In your case the 5 subjects per group were fed together. So as regards consumption there are no replicas. In fact, in the results there is no information regarding consumption.
I understand the reasons given, because they are problems that I have already faced in the past.
I could suggest repeating the experiment with 5 other subjects and put them together with these data. But the problem remains that in this experiment there are no replicas for nutrition.
Author Response
For research article
|
Response to Reviewer 2 Comments
|
||
|
1. Summary |
|
|
|
Thank you very much for taking the time to review this manuscript. Please find the detailed responses below and the corresponding revisions/corrections highlighted/in track changes in the re-submitted files.
|
||
|
2. Questions for General Evaluation |
Reviewer’s Evaluation |
Response and Revisions |
|
Does the introduction provide sufficient background and include all relevant references? |
Yes |
Improvements made |
|
Are all the cited references relevant to the research? |
Yes |
Improvements made |
|
Is the research design appropriate? |
Must be improved |
Improvements made |
|
Are the methods adequately described? |
Can be improved |
Improvements made |
|
Are the results clearly presented? |
Not applicable |
Improvements made |
|
Are the conclusions supported by the results? |
Not applicable |
Improvements made |
|
3. Point-by-point response to Comments and Suggestions for Authors |
||
|
Comments 1: To be such a statistical comparison must be based on enough replicas. In this sector they are normally used from 10-12 replicas. Up to 8 cannot be accepted no less. In your case the 5 subjects per group were fed together. So as regards consumption there are no replicas. In fact, in the results there is no information regarding consumption. I understand the reasons given, because they are problems that I have already faced in the past. I could suggest repeating the experiment with 5 other subjects and put them together with these data. But the problem remains that in this experiment there are no replicas for nutrition. |
||
|
Response 1: Thank you for pointing this out. First, we recognize that the number of replications in this trial was indeed less than optimal. However, before this experiment was conducted, we reviewed some literature related to beef quality, from which we found that the number of cattle selected by these researchers in each treatment group was 3, 4, 5 or more than 5. However, considering that the Honghe yellow cattle is a special breed in Yunnan Province, the number of healthy adult individuals available for the study is limited, especially the need for slaughter experiments, which directly limited the number of samples for our study. Therefore, for this experiment in order to investigate the effects of different energy levels on slaughter performance, carcass traits and meat quality of Honghe yellow cattle, we chose five Honghe yellow cattle as a treatment group. Secondly, there were descriptive errors in the materials and methods in the second part of the manuscript, especially lines 95-98 in the 2.2 Experimental design section: "A total of 15 Honghe yellow cattle (2 years old and 259.42 ± 23.87 kg with a body weight (BW)) were separated and distributed randomly into one of three different groups, and each group cattle was housed in an individual column (2.5 m × 4 m) with a one-way ANOVA. m) with a one-way ANOVA design." At your suggestion, we have double-checked the Materials and Methods section of the manuscript. The description of the experiment is now based on the actual situation of the experiment, as follows: the experiment was carried out in the yellow cattle farm in Yuanyang County, Honghe Prefecture, Yunnan Province, and a total of 15 Honghe yellow cattle of similar age and weight were selected and divided into 3 energy level groups with 5 replicates in each group and 1 cow in each replicate, and each cow was fed in a single pen rather than the 5 test subjects were fed together. For data analysis, we used 1 cow as the statistical unit, and the number of replicates for its cows was 5. In addition, we made changes and additions to the data analysis section of 2.6. Finally, we hope that our explanation is acceptable to you. Mention exactly where in the revised manuscript this change can be found – line 97 and lines 188-192. The references are as follows: 1.Huang Q, Wang S, Yang X, Han X, Liu Y, Khan NA, Tan Z. Effects of organic and inorganic selenium on selenium bioavailability, growth performance, antioxidant status and meat quality of a local beef cattle in China. Front Vet Sci. 2023 Apr 27;10:1171751. doi: 10.3389/fvets.2023.1171751. 2.Lin H, Zhao S, Han X, Guan W, Liu B, Chen A, Sun Y, Wang J. Effect of static magnetic field extended supercooling preservation on beef quality. Food Chem. 2022 Feb 15;370:131264. doi: 10.1016/j.foodchem.2021.131264. 3.Ueda S, Hosoda M, Kasamatsu K, Horiuchi M, Nakabayashi R, Kang B, Shinohara M, Nakanishi H, Ohto-Nakanishi T, Yamanoue M, Shirai Y. Production of Hydroxy Fatty Acids, Precursors of γ-Hexalactone, Contributes to the Characteristic Sweet Aroma of Beef. Metabolites. 2022 Apr 6;12(4):332. doi: 10.3390/metabo12040332. 4.Bulkaini B, Dahlanuddin D, Ariana T, Kisworo D, Maskur M, Mastur M. Marbling score, cholesterol, and physical-chemical content of male Bali beef fed fermented pineapple peel. J Adv Vet Anim Res. 2022 Sep 30;9(3):419-431. doi: 10.5455/javar.2022.i610. 5.Chanjula P, Wungsintaweekul J, Chiarawipa R, Phesatcha K, Suntara C, Prachumchai R, Pakdeechanuan P, Cherdthong A. Effects of Supplementing Finishing Goats with Mitragyna speciosa (Korth) Havil Leaves Powder on Growth Performance, Hematological Parameters, Carcass Composition, and Meat Quality. Animals (Basel). 2022 Jun 26;12(13):1637. doi: 10.3390/ani12131637. 6.Li L, Zhu Y, Wang X, He Y, Cao B. Effects of different dietary energy and protein levels and sex on growth performance, carcass characteristics and meat quality of F1 Angus × Chinese Xiangxi yellow cattle. J Anim Sci Biotechnol. 2014 Apr 16;5(1):21. doi: 10.1186/2049-1891-5-21.
|
||